# Molecular and Cellular Neurobiology of Circadian and Circannual Rhythms in Migraine: A Narrative Review

**DOI:** 10.3390/ijms241210092

**Published:** 2023-06-13

**Authors:** Noboru Imai

**Affiliations:** Department of Neurology and Headache Center, Japanese Red Cross Shizuoka Hospital, Shizuoka 420-0853, Japan; neurologyimai@gmail.com

**Keywords:** migraine, circadian rhythms, circannual rhythms, suprachiasmatic nucleus, PACAP

## Abstract

Migraine—a primary headache—has circadian and circannual rhythms in the onset of attacks. The circadian and circannual rhythms involve the hypothalamus, which is strongly associated with pain processing in migraines. Moreover, the role of melatonin in circadian rhythms has been implied in the pathophysiology of migraines. However, the prophylactic effect of melatonin in migraines is controversial. Calcitonin gene-related peptide (CGRP) has recently attracted attention in the pathophysiology and treatment of migraines. Pituitary adenylate cyclase-activating peptide (PACAP)—a neuropeptide identical to CGRP—is a potential therapeutic target after CGRP. PACAP is involved in the regulation of circadian entrainment to light. This review provides an overview of circadian and circannual rhythms in the hypothalamus and describes the relationship between migraines and the molecular and cellular neurobiology of circadian and circannual rhythms. Furthermore, the potential clinical applications of PACAP are presented.

## 1. Introduction

Migraine is a neurological disorder that causes recurrent headaches with intensity ranging from moderate to severe. It is characterized by pulsating pain that is often on one side of the head and aggravated by routine physical activity, and is typically accompanied by nausea, vomiting, and sensitivity to light and sound [1]. Migraine headaches are caused by a combination of genetic and environmental factors [2]. Although the exact cause of migraines is yet to be determined, it is associated with changes in the brainstem and its interactions with the trigeminal nerve—a major pain pathway [2].

The hypothalamus is a small region at the base of the brain that plays a critical role in regulating various bodily functions, including sleep, appetite, and hormone release. It also plays a role in the development of migraines [3,4]. The hypothalamus is involved in regulating the body’s circadian rhythms, the internal biological processes that control sleep and wake cycles [5,6]. Migraineurs often report that their headaches are triggered by changes in sleep patterns or irregular sleep schedules, suggesting that disruptions in the circadian rhythms may play a role in the development of migraines [7,8,9].

Circadian and circannual rhythms regulate various physiological functions, including sleep–wake cycles, hormone secretion, metabolism, and immune function [5,6,8,9,10]. These rhythms are driven by endogenous biological clocks that are located in various tissues and organs throughout the body. The central pacemaker of these rhythms is the suprachiasmatic nucleus (SCN) of the hypothalamus, which generates and synchronizes circadian rhythms in mammals.

In this review, I discuss the latest findings on the role of the hypothalamus in regulating circadian and circannual rhythms, and its association with migraines. I also discuss the potential therapeutic strategies targeting the hypothalamus for migraine management and highlight the importance of understanding the underlying biological mechanisms of this disorder.

## 2. The SCN of the Hypothalamus

The SCN controls the body’s circadian rhythms [10,11]. These rhythms are biological processes occurring on a 24 h cycle and include sleep–wake cycles, hormone production, and other bodily functions. The SCN is located in the hypothalamus, a region of the brain that regulates many bodily functions, including hunger, thirst, and temperature regulation. It is a small region of approximately 20,000 neurons located in the anterior hypothalamus, just above the optic chiasm (Figure 1) [5]. Additionally, it can be divided into two subregions—“core” and “shell”—based on their connection to the retina and output pathways, as well as the expression of specific neuropeptides. The core region contains cells that express vasoactive intestinal (VIP) and gastrin-releasing (GRP) peptides, whereas the shell region consists of cells that express arginine vasopressin (AVP) [10]. The optic chiasm is the point where the optic nerves (from the eyes) cross. This location is important for the function of the SCN because it allows it to receive information about light and darkness.

One of the most important functions of the SCN is to regulate the sleep–wake cycle [5,6,7,8,9,10,11,12,13,14,15,16], also known as the circadian rhythm. This cycle controls when humans feel sleepy and wakefulness. The SCN receives information about light and darkness through the optic nerves and uses this information to adjust the body clock. When the eyes detect light, the SCN is activated and sends signals to the pineal gland to stop producing melatonin, a hormone that induces drowsiness. The SCN is less active when it is dark. Thus, the pineal gland begins to produce melatonin, leading to drowsiness and eventual sleep.

The SCN can maintain its own circadian rhythm, even in the absence of external cues [5,10,11,12,13,14,15,16]. This ability is known as “endogenous rhythmicity” and is due to the molecular clock within the region. This clock consists of a group of proteins that interact with each other in a rhythmic manner, with the levels of each protein peaking and declining at specific times of the day. These proteins include the circadian locomotor output cycles protein kaput (CLOCK) and brain and muscle ARNT-like 1 (BMAL1; also, ARNTL), and the period (PER) and cryptochrome (CRY) proteins (Figure 1) [11]. The molecular clock regulates the expression of genes that control circadian rhythms in the body. Moreover, the regulation of circadian rhythms depends on intracellular transcription–translation negative feedback loops (TTFLs) [11,12], which have been identified in neurons and astrocytes of the SCN. The astrocytic TTFL can independently drive molecular oscillations and circadian behavior in mice, even in the absence of other cellular clocks.

The molecular clock is not unique to the SCN and is present in many cells and tissues throughout the body [5,7,11,12]. However, the molecular clock in the SCN is particularly important because it acts as a “master clock” that synchronizes rhythms in the rest of the body. The SCN sends signals to other regions of the brain and the rest of the body to regulate the timing of physiological processes such as hormone production and metabolism.

## 3. Circadian Rhythm

The circadian rhythm is a 24 h cycle of physiological and behavioral processes controlled by a molecular clock composed of clock genes and their protein products [5,6,8,9,10,11]. Its regulation is complex and involves the interaction of several neurotransmitters, hormones, and signaling pathways [5,6,7,8,9,10,11,12,13,14,15,16]. For example, the neurotransmitter serotonin plays a critical role in regulating the sleep–wake cycle and the timing of SCN activity. Serotonin is synthesized from tryptophan by the enzyme tryptophan hydroxylase and metabolized by the enzyme monoamine oxidase. Dysregulation of the serotonin system has been implicated in several sleep disorders, such as insomnia and hypersomnia, and mood disorders, such as depression and anxiety.

The molecular clock controlling the circadian rhythm is present in nearly all cells in the body. Furthermore, it is regulated by a network of transcriptional and post-transcriptional feedback loops involving the expression and degradation of clock genes and their protein products (Figure 2).

Central components of the clock network include the transcriptional activators—CLOCK (along with its paralog NPAS2) and BMAL1—which work together to stimulate the expression of the period (Per1 and Per2) and cryptochrome (Cry1 and Cry2) genes at the beginning of each cycle. As the products of Per and Cry genes accumulate, they form dimers. Additionally, they form a complex that translocates to the nucleus where it interacts with CLOCK and BMAL1 to ultimately repress its own transcription. The feedback loop occurs over approximately 24 h, and the degradation of the PER and CRY proteins is strictly controlled by E3 ubiquitin ligase complexes. In addition to the core CLOCK-BMAL1/PER-CRY feedback loop, there are other interlocking loops. One of the most notable interlocking loops involves Rev-Erbα (Nr1d1) and Rora. These two components are direct targets of CLOCK-BMAL1 and exert feedback effects on the transcription of Bmal1 (and to a lesser extent CLOCK), resulting in the antiphase oscillation of BMAL1. Other feedback loops include members of the PAR-bZip family, such as DBP, HLF, and TEF, as well as the bZip protein E4BP4 (Nfil3) and the bHLH proteins DEC1 and DEC2 (Bhlhb2 and Bhlhb3). All these components are transcriptional targets of CLOCK-BMAL1.

## 4. Circadian Rhythm

Seasonal changes related to photoperiods have been documented in humans. These changes are associated with seasonal affective disorder (SAD), a mood disorder characterized by depressive episodes in winter and remission of symptoms in summer [16]. Bright light therapy at dawn is an effective treatment for SAD, non-seasonal major depression, postpartum depression, and bipolar disorder. Furthermore, this therapy may improve the quality of life in patients with Alzheimer’s and Parkinson’s diseases [6,17,18,19,20,21]. Intrinsically photosensitive retinal ganglion cells (ipRGCs) detect non-image-forming light in mammals and are a major target of the SCN, the circadian pacemaker that synchronizes its neuronal activity to the external light–dark cycle and communicates time of day information to other brain regions [22,23,24]. The SC nuclei encode photoperiod information and create an internal representation of day length. This information is then transmitted to other brain nuclei, including the pineal gland, which regulates seasonal reproductive cycles through melatonin release [24].

The SCN regulates circadian rhythms and expresses several neurotransmitters and neuropeptides, including gamma-amino butyric acid (GABA), VIP, GRP, AVP, and neuromedin S (NMS) [6,25,26,27]. Moreover, it adapts to different photoperiods through neural plasticity and changes in clock gene expression. This can lead to changes in the organization of the SCN pacemaker and the number of neurons expressing certain neurotransmitters, such as AVP and neurotransmitters (NT), in both the SCN and paraventricular nucleus (PVN). Altered photoperiods can induce plasticity in the PVN, subsequently affecting stress response, physiology, and behavior. Therefore, seasonal changes in light exposure may contribute to the regulation of seasonal changes in physiology and behavior through photoperiod-induced plasticity in the PVN [6,28,29,30,31,32].

Exposure to short days leads to a decrease in VIP neurons and an increase in NMS-expressing neurons in mouse SCN [6]. Short days induce neurotransmitter switching in pre-existing NMS neurons, causing them to acquire the NMS phenotype. NMS neurons in the SCN travel to dopamine (DA) neurons in the PVN and regulate tyrosine hydroxylase and corticotropin-releasing hormone (CRH) neuron activity in response to photoperiods. Thus, chronic inhibition of NMS neurons during a short photoperiod delays the onset of locomotor activity and reduces the number of DA neurons in the PVN. However, selective activation of NMS neurons (but not VIP neurons) using chemogenetics under 12 h light:12 h dark (12L:12D) conditions alters the daily rhythm of locomotor activity and increases the number of DA neurons in the PVN (Figure 3) [6].

Short photoperiods increase the number of neurons expressing NMS in the SCN in adult mice. Contrastingly, long photoperiods increase the number of neurons expressing VIP. NMS and VIP co-expression is increased in SCN neurons of short-day exposed mice, consistent with neurotransmitter switching. The increase in the number of PVN DA neurons previously observed in short-day exposed mice also occur by artificially activating NMS neurons in the SCN without photoperiodic exposure. Light input to the SCN is associated with an activity-dependent mechanism of neurotransmitter switching involving the recruitment of newly expressed NMS neurons from a VIP-expressing reserve through an activity-dependent mechanism that modulates the number and activity of DA cells in the PVN. Multisynaptic neurotransmitter plasticity between different hypothalamic nuclei (PVN and SCN) induced by seasonal changes in photoperiods is an unidentified form that simultaneously mediates enhanced DA and NMS transmission and reduced CRH activity, which affects behavior. NMS/VIP co-expression is indicated by a red circle inside a blue circle. SST, somatostatin; D2R, dopamine type 2 receptor; NMUR2, NMS receptor. Data from Porcu A, et al. [6].

## 5. Circadian and Circannual Rhythms in Migraine

The potential role of circadian rhythms in the pathogenesis of migraines has been investigated, linking mutations in a component of the molecular circadian clock to migraines in animals [33]. Similarly, circadian phase delays and lower plasma melatonin levels have been reported in migraineurs [34]. Research on the prophylactic effect of melatonin in migraines has shown contrasting results. A randomized, placebo-controlled trial of 2 mg extended-release melatonin showed no prophylactic effect [35]. Contrastingly, melatonin had a considerable effect compared to a placebo and amitriptyline in a randomized trial using a placebo, 25 mg of amitriptyline, and 3 mg of melatonin (Figure 4) [36], which was better tolerated than amitriptyline. The dim light melatonin onset study showed that circadian misalignment and delayed sleep timing were associated with higher migraine frequency and severity, independent of sleep duration [37]. These findings suggest that circadian factors may play a role in the development and maintenance of chronic migraines, particularly in combination with biological or environmental factors, and that melatonin may be a target for migraine prophylaxis.

Research on the possible role of circannual rhythms in the pathogenesis of migraine is limited. In an Arctic region where light conditions are extreme, migraineurs with aura were more likely to have attacks during the light season, with light exposure being a key factor [38]. These migraineurs also reported interictal light hypersensitivity, light exposure as a precipitating factor, and frequent use of sunglasses to prevent attacks [38]. Another study showed that female migraineurs with aura had a marked seasonal variation, with more attacks during the light season [39]. A recent study using Google Trends found that interest in headaches and migraine peaked in February, October, and November and was lowest in July [40]. These studies suggest that many migraineurs may experience seasonal variations in their headaches and that further study is required in clinical practice.

## 6. PACAP and Circadian Rhythm

The circadian rhythm is regulated by the SCN through cell-autonomous transcription–translation feedback loops. Light-sensitive projections from the retinohypothalamic tract to the SCN signal via the neurotransmitters glutamate, aspartate, and PACAP [41,42,43]. Glutamate is the major neurotransmitter used for signaling [44,45]. Furthermore, PACAP is involved in the daytime regulation of the circadian cycle and contributes to the gain control mechanism for glutamate-induced phase shifts [46]. Knock-out mice deficient in PACAP or PAC1 show the stable expression of clock genes but have impaired photic entrainment and disrupted circadian food anticipation behavior, suggesting the importance of PACAP in regulating the circadian rhythm [47,48,49,50]. PACAP and glutamate induce phase changes via the light-sensitive clock genes per1 and per2 [46]. Glutamate induces per1 and per2 expression, whereas PACAP modulates their expression, inducing per1 and per2 at nanomolar concentrations and blocking glutamate-induced expression at micromolar concentrations [46].

Casein kinase 1δ (CKIδ) has been identified as an issue in migraines and circadian rhythms related to PER2 regulation. In a previous study, two families with familial migraines with aura and familial advanced sleep phase syndrome had a distinct missense mutation in the gene encoding casein kinase Iδ [51]. CK1δ regulates the rate of the mammalian circadian clock [51,52,53]. The resulting mutations (T44A and H46R) occurred in the conserved catalytic domain of CKIδ and caused reduced enzyme activity. Mice engineered to carry the CKIδ T44A allele were more sensitive to pain after treatment with nitroglycerin, a drug that triggers a migraine-like attack. In addition, they had a reduced threshold for cortical spreading depression and greater arterial dilation during cortical spreading depression. The mutation in the gene encoding CKIδ co-segregated with both the presence of migraine and advanced sleep phase [33].

## 7. PACAP and Migraine

The role of neuropeptides, particularly calcitonin gene-related peptide (CGRP) and PACAP, in the pathogenesis and treatment of migraines has received increasing attention over the past decade [54,55,56]. Monoclonal antibodies (mAbs) that block CGRP have been effective in many migraineurs. However, approximately 40% of migraineurs do not respond to this treatment.

PACAP is a member of the vasoactive intestinal peptide (VIP)/secretin/glucagon family of peptides and is widely distributed throughout the central and peripheral nervous systems. It has two isoforms—PACAP-27 and PACAP-38—which differ by six amino acids at the C-terminal (Figure 5) [56].

PACAP binds to three G protein-coupled receptors—PAC1, VPAC1, and VPAC2—which are expressed in different tissues throughout the body. PACAP-27 and PACAP-38 induce migraine attacks when administered to migraineurs [55,57,58]. An animal study showed that the peptide PACAP-38 can cause considerable hypersensitivity and dilation of the carotid arteries independent of CGRP. Unlike glyceryl trinitrate, which requires CGRP to induce hypersensitivity, PACAP-38-induced hypersensitivity is only partially inhibited by ATP-sensitive potassium channels. These findings establish the PACAP pathway as distinct from other migraine-provoking pathways, such as CGRP and glyceryl trinitrate, based on several migraine-relevant models. This understanding of PACAP signaling may lead to the identification of novel therapeutic targets, particularly for migraineurs who do not respond to anti-CGRP therapy. PACAP may be involved in non-responsiveness to CGRP mAbs, and a neutralizing mAb against PACAP is in phase II clinical development [56].

## 8. VIP

VIP is a 28 amino acid peptide known for its ability to induce vasodilation and is structurally related to PACAP and VIP (68% homology) [59,60,61,62,63,64]. VIP, PACAP, and their receptors are widely expressed in both the CNS and the peripheral tissues, particularly in neurons. In the CNS, PACAP, and VIP are released from brain neurons and bind in a paracrine manner to receptors on neighboring neurons, astrocytes, microglia and/or peripheral inflammatory cells [59,60]. PACAP and VIP interact with three G protein-coupled receptors, including VPAC1, VPAC2, and PAC1. PACAP has a high affinity for all three peptides, whereas VIP has a high affinity for VPAC1 and VPAC2 [59,60]. Whether VPAC1, VPAC2, and PAC1 receptors are present in the trigeminal ganglion and spinal trigeminal nuclei remains controversial. In an animal study, PAC1, VPAC1, and VPAC2 receptors were not stained in trigeminal ganglia, implying their absence [61]. However, PAC1 was stained in neurons innervated by PACAP nerve endings in the spinal trigeminal nucleus. Furthermore, another study found VPAC1, VPAC2, and PAC1 receptors in the trigeminal ganglion and spinal trigeminal nucleus [62]. In the latter study, high concentrations of PACAP-38 caused concentration-dependent CGRP release in the spinal trigeminal nucleus but not in the trigeminal ganglion; VIPs did not show CGRP release in either tissue.

Serum VIP is elevated not only during migraine attacks but also during interictal periods of episodic and chronic migraines [63,64]. A study investigating the efficiency of CGRP, PACAP, and VIP as migraine biomarkers using blood samples derived from chronic migraineurs, episodic migraineurs, and healthy controls revealed that VIP and PACAP could increase the risk of chronic migraines but not that of recurrent migraines. In contrast, CGRP could not predict either chronic or recurrent migraines [64].

The biomarker potential of VIP in migraines is comparable to that of PACAP; however, their involvement in migraine attacks is thought to be different. Both PACAP isoforms, PACAP-27 and PACAP-38, induce migraines [55,57,58]. Conversely, VIP, a vasodilator similar to PACAP, has a lower induction rate than PACAP [65]. In a double-blinded, crossover study in female migraineurs without aura, 73% of patients had a migraine-like attack following a PACAP38 infusion, whereas only 18% of patients had a migraine-like attack following a VIP infusion [65]. This difference in the induction of migraine-like attacks between VIP and PACAP was attributed to the non-involvement of VIP in a potent migraine-inducing pathway, unlike PACAP.

## 9. GRP

No studies have reported the role of GRP in migraines. Several studies have shown that CGP and the GRP receptors are associated with itching but not pain [66,67]. One animal study has suggested that painful stimulation results in the activation of itch-related circuits via acute pain signaling, owing to the expression of GRPs in the dorsal horn of the spinal cord [68]. However, no studies have been reported in the trigeminal ganglion or trigeminal nucleus.

## 10. AVP

Although studies on AVP in the 1980s suggested its efficacy as a therapeutic agent, no further studies were reported [69]. More recently, trigeminal ganglion neurons have been implied as a molecular fingerprint of the arginine vasopressin receptor 1A [70].

## 11. GABA

No recent studies have identified the role of GABA as a potential biomarker or therapeutic target for migraine treatment. Although associations have been reported between genetic variants related to the GABA A receptor gene cluster (GABRE, GABRA3, and GABRQ) and migraines, no association has been found in large studies. Moreover, genetic variants of GABA A receptors are considered unlikely to be a major factor in the pathogenesis of migraines [71,72,73].

## 12. NMS

Neuromedin S was isolated from the rat brain as an endogenous ligand for type 1 and type 2 neuromedin U (NMU) receptors [74]. NMU was discovered in 1985 and named for its ability to stimulate smooth muscle contraction in the uterus [74]. NMU is a highly conserved neuropeptide that exists as multiple isoforms in various species and plays roles in smooth muscle contraction, increased blood pressure, gastric emptying, and alteration of intestinal ion transport and motility. Specific NMU receptors have been identified and characterized, and NMU has been shown to be involved in the regulation of body metabolism with anorexigenic effects. The precursors of NMU and NMS have a high structural similarity; therefore, researchers are currently investigating other peptides derived from them [74]. Genome-wide association studies in African American children with migraines identified a variant at 5q33.1. This genetic variant was strongly correlated with NMUR2 mRNA expression levels [75].

## 13. Dopamine

Dopamine has been implicated in the pathogenesis of migraines for the past 30 years. Previous literature has shown that migraineurs are more sensitive to dopamine agonists for some symptoms of migraine aura, such as nausea and yawning. Nonspecific dopamine D(2) receptor antagonists are prescribed for the treatment of acute migraines [76,77].

The role of dopaminergic mechanisms in migraines is postulated to be due to the presence of dopamine receptors in the trigeminovascular system, which is thought to be responsible for headache pain, and the firing of neurons here is inhibited by dopamine agonists. Direct action of dopamine and its receptor agonists on neurons of the trigeminal cervical complex inhibits their activation after nociceptive stimulation [76]. The source of dopamine is thought to be the dopaminergic A11 nucleus of the hypothalamus. The genes for dopamine beta-hydroxylase and the dopamine transporter SLC6A3 may be involved in the pathogenesis of migraines. However, it should be noted that dopamine has little effect on peripheral trigeminal afferents [77].

A recent study reported that stress can activate kappa opioid receptors in nodal endocervical dopaminergic neurons, increase circulating prolactin, and lead to female-selective sensitization of trigeminal nociceptors through the dysregulation of prolactin receptor isoforms [78]. This study suggests that dopamine agonists and prolactin/prolactin receptor antibodies may improve the treatment of migraines and other stress-related neurological disorders in women. Based on these findings, it may be time to bring dopamine back into the spotlight.

## 14. Orexin and Circadian Rhythm

Orexins, also known as hypocretins, are neuropeptides synthesized in the lateral hypothalamus [79,80,81,82]. They are derived from the same precursor, prepro-orexin, and are modified to produce the following two peptides: orexin A (OXA), in a 33-residue peptide, and orexin B (OXB), in a 28-residue peptide [80,81]. These peptides have an excitatory effect on two G protein-coupled receptors (GPCRs) known as orexin receptor 1 (OXR1) and orexin receptor 2 (OXR2). While OXA has the same affinity for both receptors, OXB has a 10-fold higher affinity for OXR2. Orexinergic neurons have widespread projections to different areas of the brain, and the expression of receptors in these areas follows the projections. Some of these areas exclusively express OXR1, such as the locus coeruleus (LC), while others exclusively express OXR2, such as the tuberomammillary nucleus and the rostral ventromedial medulla [79,82]. The extensive projections of orexinergic neurons suggest their involvement in the modulation of homeostatic functions, including appetite, sleep–wake cycles, hormone secretion, and autonomic regulation.

The interaction between orexin and circadian rhythms is bidirectional [80]. Generally, the circadian rhythm influences the activity of orexin neurons. The master circadian clock located in the SCN of the hypothalamus sends signals to the orexin neurons to regulate their activity. This helps to synchronize the timing of orexin release with optimal periods of wakefulness during the day. However, orexin also influences the circadian rhythm. Orexin neurons receive inputs from the SCN and other brain regions involved in circadian regulation, allowing them to modulate the timing and intensity of wakefulness and sleep. Orexin promotes wakefulness and inhibits sleep by stimulating other wake-promoting centers in the brain and inhibiting sleep-promoting areas. This promotes the consolidation of wakefulness during the day and helps maintain an appropriate sleep–wake balance.

## 15. Orexin and Migraine

Orexin is involved in the modulation of pain perception and may play a role in the development and progression of migraines [79,82]. Some studies have indicated that individuals with migraines, both chronic and episodic, display dysregulation in orexin levels measured in the cerebrospinal fluid (CSF). Specifically, episodic migraineurs tend to have lower levels of orexin, while patients with chronic migraines that overuse headache medication exhibit higher levels [79,83,84].

One hypothesis is that orexin may contribute to the initiation and maintenance of migraine attacks through its effects on pain pathways [79,80,82,83,85,86,87]. Orexin receptors are found in several brain regions involved in pain processing, including the trigeminal system, which plays a critical role in migraines. Activation of orexin receptors in these areas may enhance pain transmission and sensitization, potentially leading to the development of migraine symptoms.

Migraine drugs that target orexins, specifically dual orexin receptor antagonists (DORAs), have been investigated as a potential treatment for migraines [79,88,89]. Preclinical studies have shown promising results, suggesting that DORAs may attenuate trigeminal nociceptive activity and increase thresholds for cortical spreading depression, a phenomenon associated with migraine aura. However, the clinical efficacy of DORAs is controversial, as selective targeting of individual receptors may be more effective. Differential effects of OXA and OXB on trigeminal nociception have been observed, with OXA inhibiting and OXB facilitating activation. Clinical studies are limited, with only one randomized controlled trial of filorexant, a DORA, showing no statistically significant difference in migraine or headache days compared to the placebo. However, the negative results could be attributed to the nighttime dosing and short half-life of filorexant [90]. Further investigation into selective receptor targeting and alternative dosing strategies may offer potential therapeutic benefits for migraine prophylaxis.

## 16. Serotonin (5-HT) and Circadian or Circannual Rhythms

5-HT is involved in the regulation of circadian rhythms via its role in transmitting light signals to the SCN, diurnal variation in release, effect on sleep–wake transitions, involvement in melatonin synthesis, and association with mood regulation [91,92].

5-HT is highest during the active phase in both nocturnal and diurnal rodents, suggesting an endogenously generated rhythm. The expression of the serotonin transporter and the reuptake activity of 5-HT also show time-dependent changes, with higher levels during the dark phase and lower levels during the light phase [91]. In addition, the release of the 5-HT metabolite, 5-hydroxyindoleacetic acid (5-HTIAA), follows a diurnal pattern, with greater release occurring during the dark phase. These findings suggest that 5-HT signaling is more active during the active phase.

The circadian regulation of 5-HT involves input from the sympathetic nervous system [91]. Blocking adrenergic receptors or removing the superior cervical ganglion, which is part of the sympathetic nervous system, disrupts the circadian rhythm of 5-HT. The TPH2 gene, which is responsible for 5-HT synthesis, also shows circadian variation within the raphe nuclei of the midbrain, with peak expression occurring before the light–dark transition. This rhythm appears to be influenced by the daily surge of glucocorticoids, as suppression of endogenous glucocorticoids by adrenalectomy reduces the expression of TPH2 and 5-HT, whereas replacement with exogenous glucocorticoids restores the rhythm.

5-HT is also involved in the regulation of circannual rhythms [92]. Several studies have implicated photoperiod as a risk factor for various psychiatric disorders. 5-HT and dopamine turnover, as well as SERT binding, have been observed to fluctuate over the seasons, with lower levels in winter and autumn. Furthermore, an interaction between candidate genes for affective disorders and winter births has been identified, suggesting a gene–environment risk for these disorders. These findings highlight how the photoperiod can influence the 5-HT and dopamine systems, suggesting its impact on the development and underlying mechanisms of affective disorders.

## 17. 5-HT and Migraine

5-HT plays an important role in the pathophysiology of migraines [93,94]. The relationship between 5-HT and migraines is well documented, including 5-HT imbalance, blood vessel regulation, 5-HT receptors, and pain modulation. 5-HT is involved in the progression of pain, as evidenced by the use of triptans (serotonin receptor 1B/1D (5-HT_1B/1D_) agonists) to abort acute migraine attacks. Several studies have reported altered 5-HT blood levels around and during migraine attacks. However, results have varied between studies and between individuals [93].

Recent serotonin research focused on the 5-HT_1F_ receptor agonist, lasmiditan. Lasmiditan, the only selective 5-HT_1F_ agonist approved for the treatment of acute migraine attacks, was approved in the United States in October 2019 [94]. Preclinical studies have shown that lasmiditan is non-vasoconstrictive and can penetrate the blood–brain barrier to reach the central nervous system [94]. Lasmiditan has been shown to be effective in the acute treatment of migraines in randomized, double-blind, placebo-controlled clinical trials [95,96,97,98]. Lasmiditan works by selectively activating 5-HT_1F_ receptors located in the trigeminal and central nervous systems [94]. By stimulating only the 5-HT_1F_ receptor, lasmiditan may reduce neurogenic inflammation without constricting blood vessels. Lasmiditan is indicated for the acute treatment of migraine attacks, specifically for the relief of moderate to severe headache pain. Clinical studies have shown that lasmiditan is effective in providing freedom from pain and the most troublesome symptoms such as nausea, photophobia, and phonophobia [95,96,97,98]. Common side effects of lasmiditan include dizziness, fatigue, paresthesia, and sedation, which may lead to central nervous system depression.

## 18. Sex Differences in Circadian Abnormalities and Migraine

The exact mechanisms underlying sex differences in circadian abnormalities and migraine are not fully understood. However, studies have reported sex differences in circadian regulation and sleep patterns. Adult females tend to have earlier bedtimes and wake-up times than males and are more likely to experience insomnia and sleep disorders [99,100,101,102]. These sleep-related differences may contribute to the sex difference in migraine prevalence.

In addition, sex hormones, particularly estrogen, may influence circadian regulation [103,104] Estrogen has been shown to affect the expression of clock genes, which play a critical role in maintaining circadian rhythms. Fluctuations in estrogen levels throughout the menstrual cycle, as well as during pregnancy and menopause, may contribute to changes in circadian rhythms and subsequently increase susceptibility to migraines in adult females.

The relationship between sex differences in migraine prevalence and circadian abnormalities is complex and multifactorial. While there is evidence of an association, more research is required to fully understand the underlying mechanisms and establish causality. Sex differences in migraine prevalence may not directly relate to circadian abnormalities.

## 19. Conclusions

This review provides an overview of circadian and circannual rhythms in the hypothalamus and describes the relationship between migraines and the molecular and cellular neurobiology of circadian and circannual rhythms. The role of PACAP in circadian rhythms and its potential clinical application in migraines were presented. Finally, the associations of migraines with circadian or circannual rhythm-related neuropeptides other than PACAP, such as VIP, GRP, AVP, NMS, and DA, were described. VIP is a potential migraine biomarker, but shows little promise as a therapeutic agent. GRP, AVP, GABA, and NMS have been poorly studied in migraine patients. DA antagonists are already used as an acute migraine treatment in the emergency department when standard treatments fail. Recent studies have shown the possibility of DA receptor drugs that can be used for migraine treatment. The central role of orexins in regulating homeostatic mechanisms suggests a potential role for orexin dysregulation in migraine pathophysiology. However, this study failed to provide evidence that the oral orexin receptor antagonist, filorexant, is effective for migraine prophylaxis when administered at night. This negative result may be due to the nocturnal dosing and short half-life of filorexant, and selective targeting of individual receptors may offer potential therapeutic benefits and is worthy of further investigation. 5-HT is involved in the regulation of circadian rhythms via its role in transmitting light signals to the SCN, diurnal variation in release, effect on sleep–wake transitions, involvement in melatonin synthesis, and association with mood regulation. 5-HT plays an important role in the pathophysiology of migraines. 5-HT agonists, triptan and lasmiditan, are already used for acute migraine treatment. However, their action is not believed to be related to circadian rhythms, but to pain suppression via the trigeminal nerve and/or the central nervous system. Elucidation of the molecular and cellular mechanisms of circadian and circannual rhythms in the hypothalamus may lead to the development of novel hypothalamus-targeted therapies for the treatment of migraines.

## Figures and Tables

**Figure 1 ijms-24-10092-f001:**
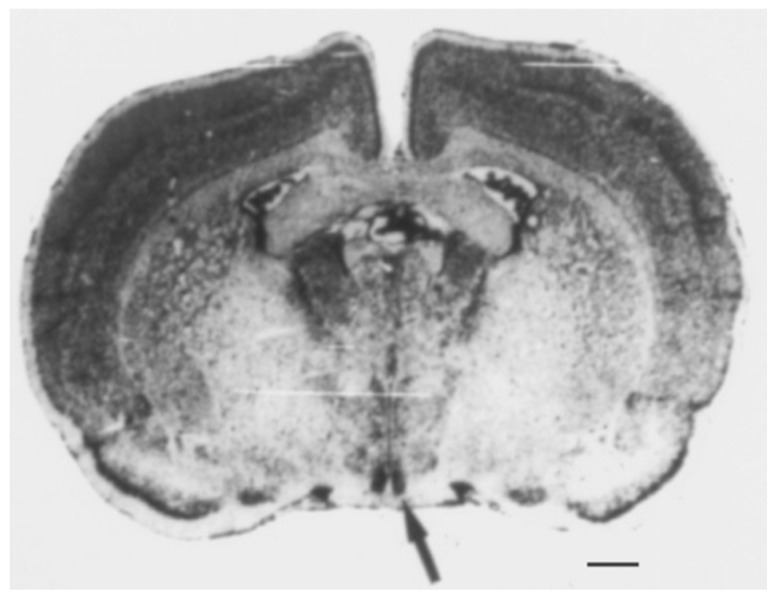
The suprachiasmatic nucleus (SCN) of the hypothalamus. The area symmetrically stained black through Nissl staining in the coronal section of the rat brain is the SCN, as indicated by the arrow. Scale bar = 1 mm. Data from Schwartz W.J. [5].

**Figure 2 ijms-24-10092-f002:**
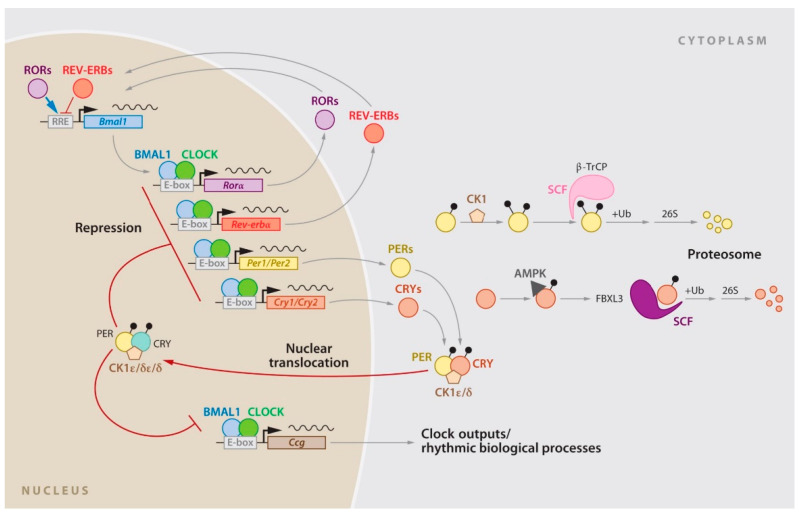
The circadian gene network in mammals. The clock network is driven by the transcriptional activators CLOCK and BMAL1, which stimulate Per1, Per2, Cry1, and Cry2 expression. The resulting proteins interact to repress their own transcription, with the stability of PER and CRY tightly regulated by parallel E3 ubiquitin ligase pathways. CLOCK and BMAL1 also regulate the nuclear receptors Rev-erbα/β, which rhythmically repress the transcription of Bmal1 and Nfil3, a process driven by the activators RORa/b. In turn, NFIL3 represses the PAR-bZip factor DBP, which regulates a rhythm in ROR nuclear receptors. These three interconnected transcriptional feedback loops are master regulators of most cycling genes. Data from Mohawk J.A, et al. [11].

**Figure 3 ijms-24-10092-f003:**
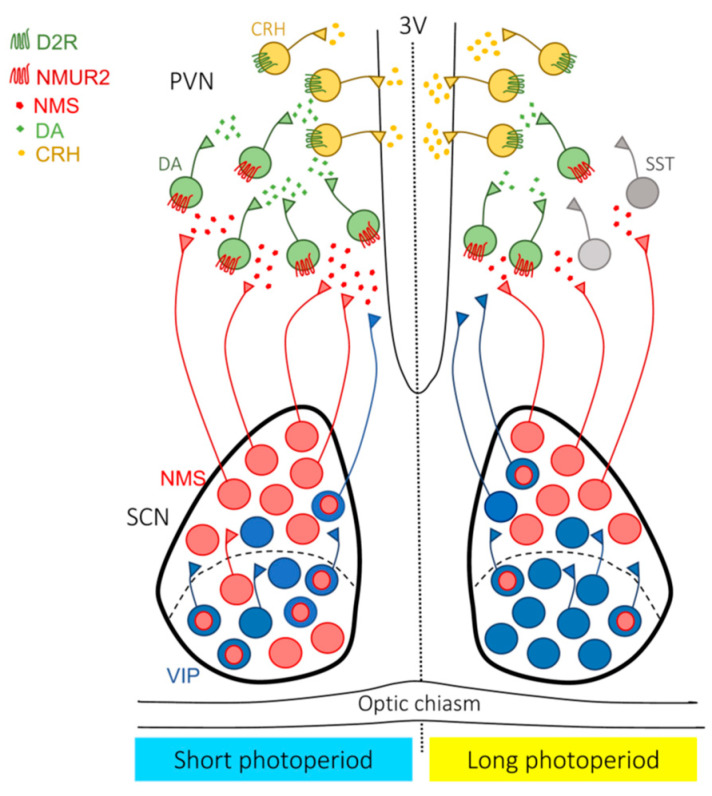
Photoperiod-induced neurotransmitter switching in the SCN-PVN network.

**Figure 4 ijms-24-10092-f004:**
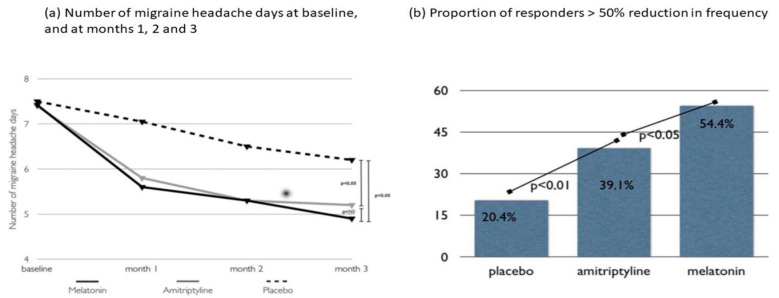
Randomized clinical trial comparing 3 mg of melatonin, 25 mg of amitriptyline, and placebo for migraine prevention. The primary efficacy endpoint of this study was the frequency of migraine headache days per month. Both melatonin and amitriptyline were superior to the placebo (*p* < 0.05) when comparing the baseline to the last month of observation. For the secondary endpoints, melatonin and amitriptyline were more effective than the placebo in reducing the number of analgesics taken as well as the duration and intensity of migraine headache attacks. Although melatonin and amitriptyline were equally effective for the primary endpoint, melatonin was superior to amitriptyline (*p* < 0.05) and the placebo (*p* < 0.01) for the secondary endpoint of the number of responders (migraineurs with a > 50% improvement in headache frequency). NS: Non-significant. Data from Gonçalves AL, et al. [36].

**Figure 5 ijms-24-10092-f005:**
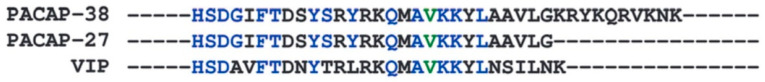
Sequence comparison of amino acids making up the isoforms of PACAP and VIP. This image shows a sequence comparison of the amino acids that make up the isoforms of PACAP (PACAP-38 and PACAP-27) and VIP. PACAP-27 is produced via post-translational truncation at the C-terminus of PACAP-38, and the amino acid sequence at the N-terminus is retained. PACAP-38 has a high degree of homology with VIP (approximately 68%). Data from Guo S, et al. [56].

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
