# Peer review of "Molecular and Cellular Neurobiology of Circadian and Circannual Rhythms in Migraine: A Narrative Review"

_ijms, 2023, doi:10.3390/ijms241210092_

Round 1
Reviewer 1 Report
The review is excellent, with little necessary comments. The authors should avoid using terms like migraineur and migraines.
It would be helpful to add clinical information such as when migraine attacks are most likely to happen given the subject.
Authors do a nice job of describing migraine targets. The should also include orexins, in light of negative studies.
The authors should also discuss gender differences in migraine prevalence and how they may or may not relate to circadian abnormalities. Clearly, the menstrual cycle ( and episodic hormone secretion) is directed by endogenous circadian activities and other biological rhythms.
minor grammatical errors such as the use of commas.
Author Response
Responses to Reviewer 1
Reviewer's report:
The review is excellent, with little necessary comments. The authors should avoid using terms like migraineur and migraines.
It would be helpful to add clinical information such as when migraine attacks are most likely to happen given the subject.
Comment 1:
Authors do a nice job of describing migraine targets. They should also include orexins, in light of negative studies.
Response:
Thank you for your comment. You have raised a valid point about the inclusion of orexins as potential targets in migraine. I have added two sections: Orexin and circadian rhythm and Orexin and migraine as follows.
- Orexin and circadian rhythm
Orexins, also known as hypocretins, are neuropeptides synthesized in the lateral hypothalamus [79-82]. They are derived from the same precursor, prepro-orexin, and are modified to produce two peptides: orexin A (OXA), in a 33-residue peptide, and orexin B (OXB), in a 28-residue peptide [80,81]. These peptides have an excitatory effect on two G protein-coupled receptors (GPCRs) known as orexin receptor 1 (OXR1) and orexin receptor 2 (OXR2). While OXA has the same affinity for both receptors, OXB has a 10-fold higher affinity for OXR2. Orexinergic neurons have widespread projections to different areas of the brain, and the expression of receptors in these areas follows the projections. Some of these areas exclusively express OXR1, such as the locus coeruleus (LC), while others exclusively express OXR2, such as the tuberomammillary nucleus and the rostral ventromedial medulla [79, 82]. The extensive projections of orexinergic neurons suggest their involvement in the modulation of homeostatic functions, including appetite, sleep-wake cycles, hormone secretion, and autonomic regulation.
The interaction between orexin and circadian rhythms is bidirectional [80]. Generally, the circadian rhythm influences the activity of orexin neurons. The master circadian clock located in the SCN of the hypothalamus sends signals to the orexin neurons to regulate their activity. This helps to synchronize the timing of orexin release with optimal periods of wakefulness during the day. However, orexin also influences the circadian rhythm. Orexin neurons receive inputs from the SCN and other brain regions involved in circadian regulation, allowing them to modulate the timing and intensity of wakefulness and sleep. Orexin promotes wakefulness and inhibits sleep by stimulating other wake-promoting centers in the brain and inhibiting sleep-promoting areas. This promotes the consolidation of wakefulness during the day and helps maintain an appropriate sleep-wake balance.
- Orexin and migraine
Orexin is involved in the modulation of pain perception and may play a role in the development and progression of migraine [79, 82]. Some studies have indicated that individuals with migraine, both chronic and episodic, display dysregulation in orexin levels measured in the cerebrospinal fluid (CSF). Specifically, episodic migraineurs tend to have lower levels of orexin, while patients with chronic migraine that overuse headache medication exhibit higher levels [79, 83, 84].
One hypothesis is that orexin may contribute to the initiation and maintenance of migraine attacks through its effects on pain pathways [79, 80, 82, 83, 85-87]. Orexin receptors are found in several brain regions involved in pain processing, including the trigeminal system, which plays a critical role in migraine. Activation of orexin receptors in these areas may enhance pain transmission and sensitization, potentially leading to the development of migraine symptoms.
Migraine drugs that target orexins, specifically dual orexin receptor antagonists (DORAs), have been investigated as a potential treatment for migraine [79, 88, 89]. Preclinical studies have shown promising results, suggesting that DORAs may attenuate trigeminal nociceptive activity and increase thresholds for cortical spreading depression, a phenomenon associated with migraine aura. However, the clinical efficacy of DORAs is controversial, as selective targeting of individual receptors may be more effective. Differential effects of OXA and OXB on trigeminal nociception have been observed, with OXA inhibiting and OXB facilitating activation. Clinical studies are limited, with only one randomized controlled trial of filorexant, a DORA, showing no statistically significant difference in migraine or headache days compared to placebo. However, the negative results could be attributed to the nighttime dosing and short half-life of filorexant [90]. Further investigation into selective receptor targeting and alternative dosing strategies may offer potential therapeutic benefits for migraine prophylaxis.

Reviewer 2 Report
The paper in object is quite clear and represents an interesting review of a niche topic. As the major point to address I would recommend to add a part on serotonin. Serotonin is indeed a key player in sleep regulation (and in sleep disorders and it represents also a key player in migraine cycle and recovery
I think that the following papers could give hints to develop a link between sleep and migraine in the discussion
Neuropsychopharmacology. 2020 Jan; 45(1): 6–20.)
Expert Rev Neurother. 2020 Mar;20(3):295-302.;
J Headache Pain. 2018; 19(1): 73.).
Author Response
Responses to Reviewer 2
Comment 1:
The paper in object is quite clear and represents an interesting review of a niche topic. As the major point to address I would recommend to add a part on serotonin. Serotonin is indeed a key player in sleep regulation (and in sleep disorders and it represents also a key player in migraine cycle and recovery
I think that the following papers could give hints to develop a link between sleep and migraine in the discussion
Arousal and sleep circuits
Barbara E. Jones
Neuropsychopharmacology. 2020 Jan; 45(1): 6–20.)
Expert Rev Neurother. 2020 Mar;20(3):295-302.;
J Headache Pain. 2018; 19(1): 73.).
Response:
Thank you for your positive feedback on the clarity and relevance of the paper, and for providing a relevant citation. I appreciate your suggestion to include a section on serotonin and its role in sleep regulation as well as its involvement in the migraine cycle and recovery.
Thus, I have added the sections: Serotonin (5-HT) and circadian or circannual rhythms and 5-HT and migraine as follows..
- Serotonin (5-HT) and circadian or circannual rhythms
5-HT is involved in the regulation of circadian rhythms via its role in transmitting light signals to the SCN, diurnal variation in release, effect on sleep-wake transitions, involvement in melatonin synthesis, and association with mood regulation [91, 92].
5-HT is highest during the active phase in both nocturnal and diurnal rodents, suggesting an endogenously generated rhythm. The expression of the serotonin transporter and the reuptake activity of 5-HT also show time-dependent changes, with higher levels during the dark phase and lower levels during the light phase [91]. In addition, the release of the 5-HT metabolite, 5-hydroxyindoleacetic acid (5-HTIAA), follows a diurnal pattern, with greater release occurring during the dark phase. These findings suggest that 5-HT signaling is more active during the active phase.
The circadian regulation of 5-HT involves input from the sympathetic nervous system [91]. Blocking adrenergic receptors or removing the superior cervical ganglion, which is part of the sympathetic nervous system, disrupts the circadian rhythm of 5-HT. The TPH2 gene, which is responsible for 5-HT synthesis, also shows circadian variation within the raphe nuclei of the midbrain, with peak expression occurring before the light-dark transition. This rhythm appears to be influenced by the daily surge of glucocorticoids, as suppression of endogenous glucocorticoids by adrenalectomy reduces the expression of TPH2 and 5-HT, whereas replacement with exogenous glucocorticoids restores the rhythm.
5-HT is also involved in the regulation of circannual rhythms [92]. Several studies have implicated photoperiod as a risk factor for various psychiatric disorders. 5-HT and dopamine turnover, as well as SERT binding, have been observed to fluctuate over the seasons, with lower levels in winter and autumn. Furthermore, an interaction between candidate genes for affective disorders and winter births has been identified, suggesting a gene-environment risk for these disorders. These findings highlight how photoperiod can influence the 5-HT and dopamine systems, suggesting its impact on the development and underlying mechanisms of affective disorders.
- 5-HT and migraine
5-HT plays an important role in the pathophysiology of migraine [93, 94]. The relationship between 5-HT and migraine is well documented, including 5-HT imbalance, blood vessel regulation, 5-HT receptors, and pain modulation. 5-HT is involved in the progression of pain, as evidenced by the use of triptans (serotonin receptor 1B/1D [5-HT1B/1D] agonists) to abort acute migraine attacks. Several studies have reported altered 5-HT blood levels around and during migraine attacks. However, results have varied between studies and between individuals [93].
Recent serotonin research focused on the 5-HT1F receptor agonist, lasmiditan. Lasmiditan, the only selective 5-HT1F agonist approved for the treatment of acute migraine attacks, was approved in the United States in October 2019 [94]. Preclinical studies have shown that lasmiditan is non-vasoconstrictive and can penetrate the blood-brain barrier to reach the central nervous system [94]. Lasmiditan has been shown to be effective in the acute treatment of migraine in randomized, double-blind, placebo-controlled clinical trials [95-98]. Lasmiditan works by selectively activating 5-HT1F receptors located in the trigeminal and central nervous systems [94]. By stimulating only the 5-HT1F receptor, lasmiditan may reduce neurogenic inflammation without constricting blood vessels. Lasmiditan is indicated for the acute treatment of migraine attacks, specifically for the relief of moderate to severe headache pain. Clinical studies have shown that lasmiditan is effective in providing freedom from pain and most troublesome symptoms such as nausea, photophobia, and phonophobia [95-98]. Common side effects of lasmiditan include dizziness, fatigue, paresthesia, and sedation, which may lead to central nervous system depression.

Reviewer 3 Report
I have reviewed the abstract, introduction, methods and materials, results, statistics, and discussion. I have also checked the references, and all appear relatively current and appropriate. Finally, I have also reviewed the figures, tables, and legends.
I find the review well-written, well-done, and informative.
Author Response
Responses to Reviewer 3
Comments and Suggestions for Authors
I have reviewed the abstract, introduction, methods and materials, results, statistics, and discussion. I have also checked the references, and all appear relatively current and appropriate. Finally, I have also reviewed the figures, tables, and legends.
I find the review well-written, well-done, and informative.
Response:
Thank you for taking the time to review the paper and for your positive feedback. I am glad to hear that you found the review to be well written, well done, and informative. Your comprehensive evaluation of each section in the manuscript is greatly appreciated.
